# Immune Checkpoint and EMT-Related Molecules in Circulating Tumor Cells (CTCs) from Triple Negative Breast Cancer Patients and Their Clinical Impact

**DOI:** 10.3390/cancers15071974

**Published:** 2023-03-25

**Authors:** Vasileios Vardas, Anastasios Tolios, Athina Christopoulou, Vassilis Georgoulias, Anastasia Xagara, Filippos Koinis, Athanasios Kotsakis, Galatea Kallergi

**Affiliations:** 1Laboratory of Biochemistry/Metastatic Signaling, Section of Genetics, Cell Biology and Development, Department of Biology, University of Patras, GR-26504 Patras, Greece; 2Oncology Unit, ST Andrews General Hospital of Patras, GR-26332 Patras, Greece; 3Hellenic Oncology Research Group (HORG), GR-11526 Athens, Greece; 4Laboratory of Oncology, Faculty of Medicine, School of Health Sciences, University of Thessaly, GR-41110 Larissa, Greece; 5Department of Medical Oncology, University General Hospital of Larissa, GR-41110 Larissa, Greece

**Keywords:** PD-L1, CTLA-4, detyrosinated α-tubulin (GLU), vimentin (VIM), circulating tumor cells, breast cancer, triple negative breast cancer, luminal

## Abstract

**Simple Summary:**

A better understanding of the molecular mechanisms that govern metastasis and the identification of early therapeutic approaches to prevent the dissemination of tumor cells in triple negative breast cancer (TNBC) patients is highly important. The present study focuses on investigating the expression of immune checkpoint molecules (PD-L1, CTLA-4) and epithelial to mesenchymal transition (EMT)-related proteins (detyrosinated α-tubulin (GLU) and vimentin (VIM)) in TNBC patients’ CTCs and assess their relations to disease severity and clinical outcome. All the examined biomarkers were found to be expressed in CTCs, whereas PD-L1, GLU, and VIM were related to worse overall survival (OS) in TNBC patients. Our data demonstrate the importance of these four biomarkers for TNBC patients and provide an interesting tool for stratifying patients that could benefit from a potential combination of novel therapies.

**Abstract:**

Triple negative breast cancer (TNBC) is the most aggressive breast cancer subtype. There are few targeted therapies for these patients, leading to an unmet need for new biomarkers. The present study aimed to investigate the expression of PD-L1, CTLA-4, GLU, and VIM in CTCs of TNBC patients. Ninety-five patients were enrolled in this study: sixty-four TNBC and thirty-one luminal. Of these patients, 60 were in the early stage, while 35 had metastatic disease. Protein expression was identified by immunofluorescence staining experiments and VyCAP analysis. All the examined proteins were upregulated in TNBC patients. The expression of the GLU^+^VIM^+^CK^+^ phenotype was higher (50%) in metastatic TNBC compared to early TNBC patients (17%) (*p* = 0.005). Among all the BC patients, a significant correlation was found between PD-L1^+^CD45^−^CK^+^ and CTLA-4^+^CD45^−^CK^+^ phenotypes (Spearman test, *p* = 0.024), implying an important role of dual inhibition in BC. Finally, the phenotypes GLU^+^VIM^+^CK^+^ and PD-L1^+^CD45^−^CK^+^ were associated with shorter OS in TNBC patients (OS: log-rank *p* = 0.048, HR = 2.9, OS: log-rank *p* < 0.001, HR = 8.7, respectively). Thus, PD-L1, CTLA-4, GLU, and VIM constitute significant biomarkers in TNBC associated with patients’ outcome, providing new therapeutic targets for this difficult breast cancer subtype.

## 1. Introduction

Breast cancer (BC) is a multifactorial disease and accounts for 30% of cancers in women [1]. Despite the latest progress in diagnostic and therapeutic approaches, breast cancer is still associated with significant morbidity and mortality. The main cause of death among BC patients is metastasis, where circulating tumor cells (CTCs) play a crucial role [2]. In fact, CTCs’ detection and enumeration have been reported in early as well as in metastatic BC and have been associated with poor clinical outcome (decreased PFS and OS) [3,4]. Epithelial markers, such as EpCAM or cytokeratins (CKs), have been widely used for the characterization of CTCs. A better understanding of the molecular mechanisms related to metastasis and the development of early therapeutic approaches to prevent the dissemination of tumor cells is highly important [5].

TNBC (ER-, PR-, HER-) represents 15% of all breast cancers and it is the most aggressive subtype, while the need for new biomarkers is urgent due to the dearth of targeted therapies available for these patients. TNBC is distinct and heterogeneous compared to the other subtypes [6]. Luminal BC is another subtype of breast cancer that is categorized into luminal A (ER+ and/or PR+, and HER2-), representing around 60% of BC and associating with a good prognosis, and luminal B (ER+ and/or PR+, and HER2+) which represents 30% of BC and is associated with high ki67 (>14%), a proliferation marker, and a poor prognosis [7].

Metastasis is associated with the presence of CTCs and disseminated tumor cells (DTCs) in peripheral blood and bone marrow, respectively [8]. Cellular mechanisms of metastasis include invasion through the stroma, escape of immune surveillance by inhibition of anti-tumorigenic processes, evasion and modulation of the tissue microenvironment, and development of resistance to therapeutic intervention [9]. During metastasis, CTCs undergo epithelial to mesenchymal transition (EMT), losing their epithelial properties and acquiring a mesenchymal phenotype with an enhanced invasive character giving them the ability to survive in the bloodstream and finally metastasize [10]. Upregulation and downregulation of specific markers take place during EMT [10]. Epithelial markers such as EpCAM and E-cadherin (E-cad) are downregulated, while mesenchymal markers such as N-cadherin (N-cad), fibronectin, matrix metalloproteinases, integrins α_v_ and β_1_, and smooth muscle actin are upregulated [10]. During EMT, transcriptional factors, such as Twist and Snail, also demonstrate significant involvement in the survival and stability of CTCs in the bloodstream, as well as the cytoplasmic protein vimentin (VIM) whose expression levels are elevated in CTCs during EMT [10]. Furthermore, tumor cells with stem cell-like characteristics, known as cancer stem cells (CSCs) have been found to display enhanced tumorigenicity, metastatic ability, and resistance to radiation and chemotherapy [11,12]. The acquisition of stem cell properties is associated with EMT [13]. Previous studies have shown that the ADH/ALDH activities are lower in tumor cells than in normal parenchyma, suggesting that isoenzymes of ADH may play an important role in carcinogenesis [14]. Among all tested classes of ADH isoenzymes, only class I had higher activity in the serum of patients with breast cancer in stage IV [15]. Furthermore, the CD44(+)/CD24(−/low) and ALDH1(+) cell phenotypes have been associated with stemness and enhanced tumorigenic potential in breast cancer and the existence of a subpopulation of CTCs with putative stem cell phenotypes in patients with metastatic breast cancer has recently been reported [16].

We have recently shown that VIM, a type III intermediate filament protein, is overexpressed in BC patients’ CTCs and its expression level is higher in the metastatic setting compared to early BC [8,17]. Furthermore, its overexpression is associated with increased invasive potential and poor prognosis [8]. VIM expression in primary tumors has been significantly correlated with reduced survival in TNBC patients. More specifically, increased VIM expression was linked to high invasiveness in TNBC patients and their poor prognosis [18]. Another marker related to EMT which is overexpressed in BC is detyrosinated α-tubulin (GLU) [8]. Tubulin detyrosination is a common event in breast cancer, easy to detect, and correlated with tumor aggressiveness [19]. GLU levels are higher in metastatic patients compared to early BC patients [8]. Additionally, GLU participates in the formation of microtentacles, which are cytoskeletal structures, constitute an important mechanism for metastatic dissemination, and are associated with EMT pathways [8,20].

Immune checkpoint molecules work as protective factors for the body’s immune system, and their role is to regulate the immune system to avoid autoimmune responses caused by excessively activated immune cells [21]. In case of overexpression or overactivation of these molecules, immune function is inhibited. Tumor cells can exploit this event by excessively activating immune checkpoint molecules to prevent local immune cells from escaping surveillance and clearance, thus accelerating tumor growth [21]. Programmed death ligand 1 (PD-L1), which is an immune checkpoint molecule, is expressed in tumor cells and interacts with programmed cell death protein 1 (PD-1) on the surface of immune cells negatively regulating the immune system [22]. An association of poor prognosis in patients with advanced non-small cell lung cancer (NSCLC) has been indicated [23]. In general, elevated levels of PD-L1 in tumor cells have been linked with poor prognosis in different tumor types, such as breast cancer, head/neck cancer, gastric cancer, and prostate cancer [24,25,26,27,28,29]. Atezolizumab, avelumab, durvalumab, and cemiplimab constitute four monoclonal anti-PD-L1 antibodies that are approved by the FDA for the treatment of different types of cancer [30]. In addition, there are FDA-approved anti-PD-1 antibodies such as nivolumab, pembrolizumab, and dostarlimab for the treatment of different types of cancer, including melanoma, renal cell carcinoma, squamous lung cancer, and metastatic NSCLC [31,32,33]. Interestingly, recent studies have demonstrated that dual inhibitors, combining anti-PD1/PD-L1 antibodies with cytotoxic agents such as tubulin inhibitors, accomplish synergistic effects and show better anti-tumor efficacy than chemotherapy alone [34].

Cytotoxic T lymphocyte-associated antigen-4 (CTLA-4) is considered the “leader” of the immune checkpoint inhibitors, as it stops potentially autoreactive T-cells at the initial stage of naive T-cell activation, typically in lymph nodes [35]. CTLA-4 is poorly understood in the BC context, and the clinical impact of CTLA-4 expression on BC treatment is still not clear [35]. There have been some studies reporting an association of its increased levels with advanced disease clinical stage, emphasizing CTLA-4’s importance in the development and progression of breast cancer [35]. Thus, CTLA-4 expression in BC can be considered a potential prognostic biomarker as well as a possible therapeutic target in the emerging field of BC immunotherapy [35]. Immunotherapies targeting PD-1/ PD-L1 and CTLA-4/B7 pathways have been shown to be clinically efficient against various cancer types [30]. An extensive summary of PD-L1 and CTLA-4 molecules and their common regulatory mechanisms could be significant for the identification of patients with favorable responses to anti-PD-L1 and anti-CTLA-4 treatments or even possible targeting of these two with dual inhibitors. Dual inhibitors represent a novel and promising approach that can show synergistic and efficient effects compared to chemotherapy [34].

The expression of all these molecules has not been elucidated to date in CTCs from TNBC. Therefore, the aim of the present study was to investigate PD-L1, CTLA-4, GLU, and VIM in CTCs of TNBC patients and to assess their relationship with the severity of disease and clinical outcome.

## 2. Materials and Methods

### 2.1. Patients’ Samples and Cytospins’ Preparation

Ninety-five blood samples were obtained from BC patients (sixty-four with TNBC and thirty-one with luminal A or B). Patients were excluded from the study if they met any of the following criteria: a. Age < 18 years, b. Patients without histologically and cytologically confirmed primary breast cancer, c. Patients who had received at least one cycle of therapy, and d. Patients without a signed informed consent form. All patients gave their written informed consent, and the study was approved by the Ethics and Scientific Committees of our institution (15/12/21-6734). Patients’ characteristics are presented in Table 1. The primary endpoint of the study was the identification of CTCs belonging to the examined phenotypes. The secondary endpoint was the investigation of the potential clinical relevance of the observed biomarkers for TNBC patients’ outcome. Peripheral blood (10 mL in EDTA) was obtained by venipuncture after discarding the first 5 mL to avoid contamination from skin epithelial cells during sample collection. Peripheral blood mononuclear cells (PBMCs) were isolated by Ficoll–Hypaque density, after centrifugation at 1800 rpm for 30 min at 4 °C. PBMCs were washed twice with PBS and centrifuged at 1500 rpm for 10 min. Aliquots of 500,000 cells were centrifuged at 2000 rpm for 2 min on glass slides [17,36]. Cytospins were dried up and stored at −80 °C. In this study, we did not use any magnetic isolation with EpCAM beads because the use of two different epithelial markers (EpCAM and CK) would decrease the recovery rate of CTCs. In addition, it has been shown that EpCAM is downregulated in many CTCs [37]. Instead, we followed the published methodology used in the past in many publications of our team [17,38,39,40].

### 2.2. Cell Cultures

Control experiments were performed using MDA-MB-231 cells as controls for TNBC cells and MCF7 cells as controls for luminal cells (both cell lines were obtained from the American Type Culture Collection, ATCC; Manassas, VA, USA). The MDA-MB-231 (basal-like) cells were cultured in high-glucose Dulbecco’s modified Eagle medium (DMEM) with 10% fetal bovine serum (FBS) and 2 mM L-glutamine (Thermo Fisher Scientific, Waltham, MA, USA). The MCF7 mammary adenocarcinoma cells were cultured in (*v*/*v*) 1:1 DMEM (Thermo Fisher Scientific, Waltham, MA, USA) supplemented with 10% FBS (Thermo Fisher Scientific), 2mM L-glutamine (Thermo Fisher Scientific), 30mM NaHCOB_3B_, 16 ng/mL insulin, and 50 mg/mL penicillin/streptomycin (Thermo Fisher Scientific). Cells were maintained in a humidified atmosphere of 5% COB_2B_–95% air. Subcultivation was performed with 0.25% trypsin and 5mM EDTA.

### 2.3. Triple Immunofluorescence

Triple immunofluorescence staining experiments for CK/CD45/PD-L1, CK/CD45/CTLA-4, and CK/GLU/VIM were performed on patients’ cytospins, accompanied by control experiments (Appendix A) on cytospins with MDA-MB-231 cells for TNBC cells and MCF7 for luminal cells. Negative controls were used by omitting the corresponding primary antibody, while including its secondary IgG antibody. The cytomorphological criteria proposed by Meng et al. [41] (for example, high nuclear/cytoplasmic ratio, larger cells than white blood cells) were used to characterize a CK-positive cell as a CTC. Cells were initially fixed with cold acetone: methanol 9:1, followed by blocking with 5% FBS overnight. PD-L1 (1:100; Novus Biologicals, Englewood, CO, USA) was detected using anti-goat antibody labeled with Alexa Fluor 488 (ThermoFisher Scientific, Waltham, MA, USA). CTLA-4 was detected using CTLA-4 (F-8) Alexa Fluor 488 (1:100; Santa Cruz Biotechnology, Santa Cruz, CA, USA). CD45 (common leukocyte antigen) antibody, which was used to exclude possible ectopic expression of cytokeratins by hematopoietic cells, was detected using anti-CD45 conjugated with Alexa Fluor 647 (1:100; Novus Biologicals, Englewood, CO, USA). GLU (1:400; Abcam, Cambridge, MA, USA) was detected using anti-rabbit antibody labeled with Alexa Fluor 647 (ThermoFisher Scientific). VIM (1:100; Santacruz) was detected using anti-goat antibody labeled with Alexa Fluor 555 (ThermoFisher Scientific). CK detection was achieved with the primary A45-B/B3 antibody (1:100; Amgen, Thousand Oaks, CA, USA) and its secondary anti-mouse antibody labeled with Alexa Fluor 488 (ThermoFisher Scientific). Finally, cells were stained with 4′,6-diamidino-2-phenylindole (DAPI)-containing anti-fade reagent. Cytospins were analyzed with the VyCAP system (VyCAP B.V., Enschede, the Netherlands) and a Leica TCS SP8 confocal microscope (Leica Microsystems, Wetzlar, Germany).

### 2.4. Statistical Analysis

Spearman analysis, a Mann–Whitney test, and χ^2^ tests were used to compare the two subtypes of cancer and the status of disease with the expression of PD-L1, CTLA-4, GLU, and VIM, as well as available clinical data. Kaplan–Meier survival tests for OS and PFS were performed. All analyses were performed on IBM SPSS statistics version 27 software (IBM, Armonk, NY, USA). A value of *p* ≤ 0.05 was used to identify significant results.

## 3. Results

### 3.1. Expression of Immune Checkpoints PD-L1 and CTLA-4 in the CTCs of TNBC Patients

Sixty-four TNBC patients were analyzed in the current study (Figure 1A). Of the TNBC patients, 50% (32 out of 64) were CK-positive (Figure 1B). Thirty-one hormone receptor-positive (luminal subtype) patients were also enrolled in this study (eighteen early and thirteen metastatic; Figure 1A). Of luminal patients, 39% (12 out of 31) were CK-positive (Figure 1B). Furthermore, 45% (19 out of 42) of early TNBC and 28% (5 out of 18) of early luminal patients were CK-positive, whereas 55% (12 out of 22) of metastatic TNBC and 54% (7 out of 13) of metastatic luminal patients were CK-positive (Figure 1C).

Among TNBC patients, the phenotypes of PD-L1^+^CD45^−^CK^+^ and CTLA-4^+^CD45^−^CK^+^ were present in 41% (26 out 64) and 36% (23 out of 64), respectively (Figure 2A). Regarding disease status, the phenotype of PD-L1^+^CD45^−^CK^+^ was present in 38% (16 out 42) of early TNBC vs. 45% (10 out of 22) of metastatic TNBC patients, while the phenotype CTLA-4^+^CD45^−^CK^+^ was present in 26% (11 out of 42) of early TNBC vs. 50% (11 out of 22) of metastatic TNBC patients (*p* = 0.025, Figure 2B, Appendix A). Hence, expression of the phenotype CTLA-4^+^CD45^−^CK^+^ was higher in metastatic compared to early TNBC patients. The frequencies of PD-L1^+^CD45^−^CK^+^ and CTLA-4^+^CD45^−^CK^+^ phenotypes among the total number of TNBC patients’ CTCs were 78% and 69%, respectively (Figure 2C).

Considering disease status, the percentage of CTCs expressing the phenotype PD-L1^+^CD45^−^CK^+^ in early vs. metastatic disease was 70% vs. 91%, respectively, whereas the percentage of PD-L1-negative CTCs was 30% vs. 9% (*p* = 0.049), respectively. The same percentages for the CTLA-4^+^CD45^−^CK^+^ phenotype were 61% vs. 80% (*p* = 0.024), respectively, while for CTLA-4^−^CD45^−^CK^+^ phenotype they were 39% vs. 20%, respectively (Figure 2D, Appendix A). Hence, frequency of the phenotype CTLA-4^+^CD45^−^CK^+^ was higher in metastatic TNBC compared to early TNBC patients’ CTCs, whereas the phenotype PD-L1^−^CD45^−^CK^+^ was higher in early TNBC patients’ CTCs. Representative images of all the different phenotypes of CTCs are illustrated (Figure 3, Appendix A).

### 3.2. Expression of Immune Checkpoints PD-L1 and CTLA-4 in the CTCs of Luminal BC Patients Compared to TNBC Patients

To explore potential differences in the expression pattern of the examined immune checkpoint molecules between luminal and TNBC patients, thirty-one patients with luminal subtypes were also examined for the expression of immune checkpoint molecules. Among BC subtypes, the phenotype of PD-L1^+^CD45^−^CK^+^ was present in 41% (26 out 64) of TNBC vs. 29% (9 out of 31) of luminal patients, while the phenotype of CTLA-4^+^CD45^−^CK^+^ was present in 36% (23 out of 64) of TNBC and 23% (7 out of 31) of luminal patients (Figure 4A and Appendix A). Furthermore, the frequency of the PD-L1^+^CD45^−^CK^+^ phenotype was 78% vs. 65% among TNBC and luminal patients’ CTCs, respectively, while the frequency of the CTLA-4^+^CD45^−^CK^+^ phenotype was 69% vs. 59% among TNBC and luminal patients’ CTCs, respectively (Figure 4C). Consequently, expression of PD-L1 and CTLA-4 was higher in TNBC compared to luminal patients, although the differences do not reach statistical significance (Appendix A).

Concentrating on the status of the disease, the PD-L1^+^CD45^−^CK^+^ phenotype was present in 38% (16 out of 42) of early and 45% (10 out of 22) of metastatic TNBC patients, while it was present in 17% (3 out of 18) of early and 46% (6 out of 13) of metastatic luminal patients (Figure 4B and Appendix A).

On the other hand, the CTLA-4^+^CD45^−^CK^+^ phenotype was present in 26% (11 out of 42) of early and 50% (11 out of 22) of metastatic TNBC patients (*p* = 0.025), while it was present in 17% (3 out of 18) of early and 31% (4 out of 13) of metastatic luminal patients (Figure 4B and Appendix A). Therefore, expression of CTLA-4 was higher in metastatic compared to early patients with both subtypes.

The frequency of the PD-L1^+^CD45^−^CK^+^ phenotype was 70% vs. 91% among early and metastatic TNBC patients’ CTCs, respectively. In terms of the luminal patients, 60% of the CTCs of early vs. 70% of the CTCs of metastatic patients belong to the PD-L1^+^CD45^−^CK^+^ phenotype (Figure 4D, Appendix A). Similarly, frequency of the CTLA-4^+^CD45^−^CK^+^ phenotype was 61% vs. 80% among early and metastatic TNBC patients, respectively, while it was 75% vs. 40% in early vs. metastatic luminal patients, respectively (Figure 4D, Appendix A).

Furthermore, significant correlation was found between PD-L1^+^CD45^−^CK^+^ and CTLA-4^+^CD45^−^CK^+^ phenotypes among all the BC patients (Spearman test, *p* = 0.024).

### 3.3. Expression of EMT-Related Molecules; GLU and VIM in the CTCs of TNBC Patients

The phenotypes GLU^+^VIM^+^CK^+^, GLU^−^VIM^+^CK^+^, GLU^+^VIM^−^CK^+^, and GLU^−^VIM^−^CK^+^ were present in 28% (18 out of 64), 44% (28 out of 64), 19% (12 out of 64), and 19% (12 out of 64) of TNBC patients, respectively (Figure 5A). Regarding the percentages of CTCs belonging to the above phenotypes among TNBC patients were: 23%, 53%, 11%, and 13%, respectively (Figure 5C).

Concerning disease status, the phenotypes GLU^+^VIM^+^CK^+^, GLU^−^VIM^+^CK^+^, GLU^+^VIM^−^CK^+^, and GLU^−^VIM^−^CK^+^ were present in 17% (7 out of 42), 43% (18 out of 42), 21% (9 out of 42), and 17% (7 out of 42) of early TNBC patients, respectively, compared to 50% (11 out of 22) (*p* = 0.005), 45% (10 out of 22), 14% (3 out of 22), and 23% (5 out of 22) of metastatic TNBC patients, respectively (Figure 5B, Appendix A). Hence, expression of the phenotype GLU^+^VIM^+^CK^+^ was higher in metastatic TNBC compared to early TNBC patients.

The frequencies of GLU^+^VIM^+^CK^+^, GLU^−^VIM^+^CK^+^, GLU^+^VIM^−^CK^+^, and GLU^−^VIM^−^CK^+^ phenotypes among CTCs in early vs. metastatic patients were 14% vs. 37% (*p* = 0.020), 57% vs. 47%, 14% vs. 7%, and 15% vs. 9%, respectively (Figure 5D, Appendix A). Hence, the frequency of the GLU^+^VIM^+^CK^+^ phenotype was higher in metastatic TNBC compared to early TNBC patients’ CTCs. Representative images of the different phenotypes are illustrated in Figure 3.

Among TNBC patients, a significant correlation was found between PD-L1^+^CD45^−^CK^+^ and GLU^+^VIM^−^CK^+^ phenotypes (Spearman test, *p* = 0.007).

### 3.4. PD-L1, CTLA-4, GLU, and VIM in the CTCs of BC Patients and Clinical Outcome

Clinical data regarding follow-up were available for 60 patients (48 out of 64 TNBC and 12 out of 31 luminal). After a median follow-up period of 7 months for TNBC and 53 months for luminal patients, 13 deaths and 1 death, respectively, were registered due to disease progression.

Among TNBC patients, the phenotype GLU^+^VIM^+^CK^+^ was associated with a shorter OS (27.1 vs. 57.1 months; log-rank *p* = 0.048, HR = 2.9; Figure 6A).

Furthermore, among TNBC patients, there was also an association of the phenotype PD-L1^+^CD45^−^CK^+^ with a shorter OS (7.6 vs. 53.8 months; log-rank *p* < 0.001, HR = 8.7; Figure 6B). More specifically, TNBC patients with two or more PD-L1+ CTCs had shorter OS than those with one or no PD-L1+ CTCs. Similarly, early TNBC patients with two or more PD-L1+ CTCs had shorter OS (log-rank *p* < 0.001, HR = 1; Figure 6C).

Among all BC patients, as expected, the metastatic ones were associated with a shorter OS (log-rank *p* < 0.001, HR = 9.7).

## 4. Discussion

In the present study, we characterized the CTCs from TNBC patients regarding the immune checkpoint antigens PD-L1 and CTLA-4 plus the expression of the EMT-related molecules GLU and VIM in CTCs of TNBC patients. In addition, we compared the expression of PD-L1 and CTLA-4 in CTCs of TNBC vs. luminal patients. We also examined the correlation of all these molecules with the severity of the disease and clinical outcome.

Based on the disease status, 17% vs. 50% of patients with early and metastatic TNBC, respectively, expressed the phenotype GLU^+^VIM^+^CK^+^ (*p* = 0.005). Therefore, the expression of this phenotype was higher in the metastatic setting. This is also supported by another study from our group including mainly luminal patients, where expression levels of GLU and VIM were higher in metastatic BC patients and this expression was related to patients’ outcome [8]. Tubulin detyrosination, which results from the suppression of tubulin tyrosine ligase and the resulting unbalanced activity of tubulin-carboxypeptidase, apparently represents a strong selective advantage for cancer cells and has been linked to poor prognosis in BC [42]. Interestingly, our findings indicated that the phenotype GLU^+^VIM^+^CK^+^ was associated with shorter OS of TNBC patients (log-rank *p* = 0.048). This finding is also similar to the results observed in NSCLC patients, where the OS has been shown to significantly decrease in patients with high GLU (3.8 vs. 7.9 months; *p* = 0.018) and/or high VIM (3.2 vs. 7.1 months; *p* = 0.029) expression in their CTCs [43]. Furthermore, the detection of GLU^+^VIM^+^CK^+^ tumor cells in peripheral blood has been correlated to shorter progression free survival (PFS) of BC patients [8]. Overexpression of GLU and VIM has been shown in cytoskeletal structures called microtentacles, which represent an important mechanism for metastatic dissemination and are associated with EMT pathways [8]. The findings of that study showed that CTCs could interact with one another through filamentous protrusions that were supported by TUB, VIM, and GLU [8]. Previous research has revealed that invasive breast carcinomas expressing vimentin have a higher rate of tubulin microtentacles after detachment than non-invasive cell lines that do not express vimentin [44]. Additionally, it has been demonstrated that GLU levels significantly rise after epithelial cell detachment, and the microtentacles that are formed due to detachment are enriched in GLU [20]. The preservation of microtentacles after detachment suggests that vimentin is aligned with GLU in microtentacles, while cytokeratin is not [44]. Based on the above evidence, GLU and VIM are associated with disease progression and constitute important biomarkers in TN breast cancer.

One of the main objectives of this study was also to evaluate the expression of immune checkpoint molecules such as PD-L1 and CTLA-4 in TNBC and luminal patients and associate these results with their clinical outcome. Among BC subtypes, the phenotype of PD-L1^+^CD45^−^CK^+^ was present in 41% of TNBC and 29% of luminal patients, while the phenotype of CTLA-4^+^CD45^−^CK^+^ was present in 36% of TNBC and 23% of luminal patients, implying that immunosuppressive phenotypes potentially predominate in TN breast cancer. In addition, the percentage of the PD-L1^+^CD45^−^CK^+^ phenotype was 78% vs. 65% among TNBC and luminal patients’ CTCs, respectively, while the percentage of the CTLA-4^+^CD45^−^CK^+^ phenotype was 69% vs. 59% among TNBC and luminal patients’ CTCs, respectively. Hence, the expression of PD-L1 and CTLA-4 seemed to be higher in TNBC compared to luminal patients, possibly linked to the higher aggressiveness of the disease, although this finding did not reach statistical significance. In agreement with these findings, it has been shown that CTLA-4 level was significantly higher in TNBC than in the luminal subtype and HER2+ subtype (*p* = 0.019 and *p* < 0.001, separately), and was significantly higher in ER- and PR-negative samples than in ER- and PR-positive samples (*p* < 0.001) [45]. Similarly, it has been demonstrated that PD-L1 expression levels are higher in TNBC than any other breast cancer subtype [46].

Based on the status of the disease, the percentage of the PD-L1^+^CD45^−^CK^+^ phenotype was 70% vs. 91% among early and metastatic TNBC patients’ CTCs, respectively, while in luminal patients the same percentages were 60% vs. 70%, respectively (Figure 4D). The CTLA-4^+^CD45^−^CK^+^ phenotype was present in 26% of early vs. 50% of metastatic TNBC patients (*p* = 0.025), while it was present in 17% of early vs. 31% of metastatic luminal patients (Figure 4B). These findings indicate a slightly higher prevalence of both immune checkpoint phenotypes (PD-L1^+^CD45^−^CK^+^ and CTLA-4^+^CD45^−^CK^+^) in metastatic settings of both TNBC and luminal patients, implying that these CTCs during disease evolution possess an increased capacity to suppress the immune system and therefore represent potential suitable targets for anti-PDL1 and anti-CTLA-4 therapies. Similar findings demonstrated significant evidence that PD-L1 was frequently expressed on metastatic cells circulating in the blood of HR^(+)^ and HER2^(−)^ breast cancer patients, whereas other studies have suggested that higher levels of CTLA-4 may be linked to more advanced stages of breast cancer, highlighting the significance of CTLA-4 in the growth and progress of the disease [24,35]. Interestingly, the phenotype PD-L1^+^CD45^−^CK^+^ was associated with shorter OS of TNBC patients. Specifically, all TNBC patients, including early TNBC patients with two or more PD-L1+ CTCs, had shorter OS (log-rank *p* = 0.048 and *p* < 0.001 log-rank respectively) than those with one or no PD-L1+ CTCs. This finding is consistent with other studies which have indicated that high levels of PD-L1 expression are linked to a significantly worse overall survival rate in breast cancer and NSCLC patients [43,47,48,49]. Additionally, it has been demonstrated that patients with metastatic breast cancer (MBC) harboring PD-L1^+^ CTCs had a shorter PFS, indicating that PD-L1 expression in MBC acts as a negative prognostic biomarker and highlighting the importance of this as a biomarker [50]. It is also noteworthy that the poorer OS (*p* < 0.001) was also confirmed in early TNBC patients, implying that PD-L1 could be a useful biomarker in the early stages of the disease for the TNBC subtype.

Among TNBC patients, significant correlation was found between PD-L1^+^CD45^−^CK^+^ and GLU^+^VIM^−^CK^+^ phenotypes (Spearman test, *p* = 0.007). This possibly explains the efficiency of the new dual inhibitors targeting both tubulin and the PD-1/PD-L1 axis. More specifically, a series of novel CA-4 analogs as dual inhibitors of tubulin polymerization and PD-1/PD-L1 were designed, synthesized, and bioevaluated. Among these CA-4 analogs, TP5 was the most effective as it exhibited moderate anti-PD-1/PD-L1 activity, while it could also effectively inhibit tubulin polymerization, suppressing HepG2 cell migration and colony formation, causing cell arrest at G2/M phase and inducing apoptosis [34]. Dual inhibitors or multitargeting drugs possess the prospect to complement or even replace chemotherapy or therapeutic regimens based on drug combinations, as they can show either additive or synergistic effects and guarantee the simultaneous presence of the molecule at the sites of action as well as interaction with its multiple targets [51].

Notably, a significant correlation was also found between PD-L1^+^CD45^−^CK^+^ and CTLA-4^+^CD45^−^CK^+^ phenotypes among all the BC patients (Spearman test, *p* = 0.024). This finding demonstrates a connection between CTLA-4/B7 and PD1/PD-L1 pathways, the two most representative immune checkpoint molecules, which negatively regulate T-cell immune function during different phases of T-cell activation [52]. Targeting checkpoints of immune cell activation has been demonstrated to be the most effective approach for the activation of anti-tumor immune responses [30]. Inhibitors targeting these pathways have revolutionized immunotherapies for several cancer types. Ipilimumab, an FDA-approved antibody targeting CTLA-4, seems to restore tumor immunity at the priming phase, whereas anti-PD-1/PD-L1 antibodies, such as pembrolizumab and nivolumab, restore immune function in the tumor microenvironment [53]. A novel and really promising approach is the use of dual inhibitors which can show synergistic and efficient effects compared to chemotherapy. An example is the development of indole alkaloid-type dual immune checkpoint inhibitors against CTLA-4 and PD-L1 based on diversity-enhanced extracts which were indicated to suppress not only CTLA-4 and PD-L1 gene expression but also protein expression on the cell surface [54].

Our study highlighted the importance of PD-L1, CTLA-4, GLU, and VIM as biomarkers in TNBC, the correlation among them, and their potential clinical significance. Furthermore, our analysis provided evidence that supports the necessity of novel and prospective dual inhibitors targeting both tubulin and the PD-1/PD-L1 axis as well as dual inhibitors against CTLA-4 and PD-L1. These findings could be of clinical importance in preventing the dissemination of cancer cells in the bloodstream.

## 5. Conclusions

TNBC is the most aggressive subtype and there is a limitation in the targeted therapeutic approaches for these patients. The field of liquid biopsy can be very beneficial in terms of stratifying patients in real time and identifying subpopulations that can benefit from specific treatments. To the best of our knowledge, this is the first study focusing simultaneously on PD-L1, CTLA-4, GLU, and VIM expression in CTCs of TNBC patients. The findings of this study demonstrate the importance of these four biomarkers which are shown to be relevant to TNBC patients’ outcomes and provide an interesting tool for stratifying patients that could benefit from the novel and prospective dual inhibitors for therapy.

## Figures and Tables

**Figure 1 cancers-15-01974-f001:**
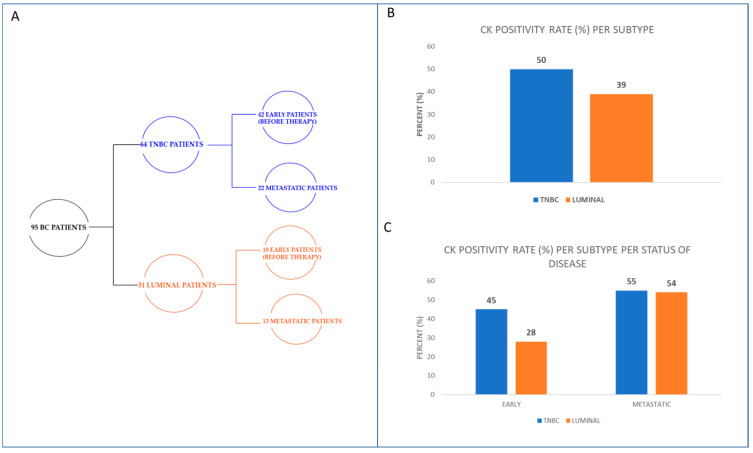
Study design and CTCs’ positivity rate in TNBC and luminal BC patients. (**A**) Flowchart showing the total number of BC patients categorized in the two major subtypes (luminal and TNBC) and their disease status (early and metastatic). (**B**) Percentage of CK+ TNBC and luminal patients. (**C**) Percentage of CK+ early and metastatic TNBC and luminal patients.

**Figure 2 cancers-15-01974-f002:**
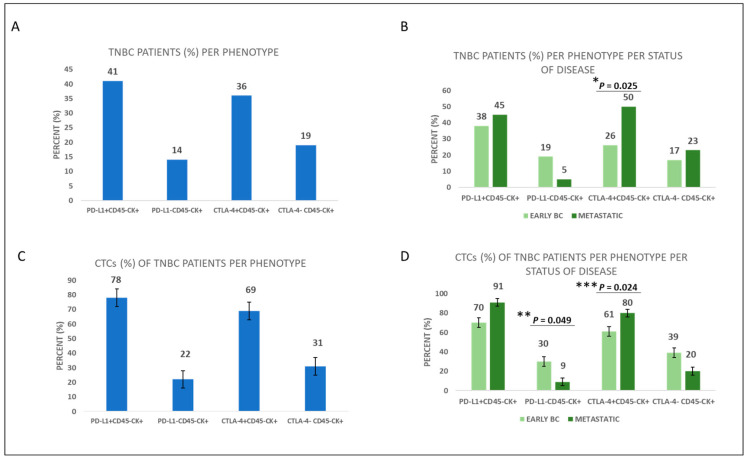
PD-L1 and CTLA-4 expression in TNBC patients’ CTCs. (**A**) Percentage of TNBC patients with all the different phenotypes. (**B**) Percentage of early and metastatic TNBC patients with the corresponding CTC phenotypes (* *p* = 0.025). (**C**) Percentage of CTCs with all the corresponding phenotypes in TNBC patients. (**D**) Percentage of CTCs with all the corresponding phenotypes in early and metastatic TNBC patients (** *p* = 0.049, *** *p* = 0.024).

**Figure 3 cancers-15-01974-f003:**
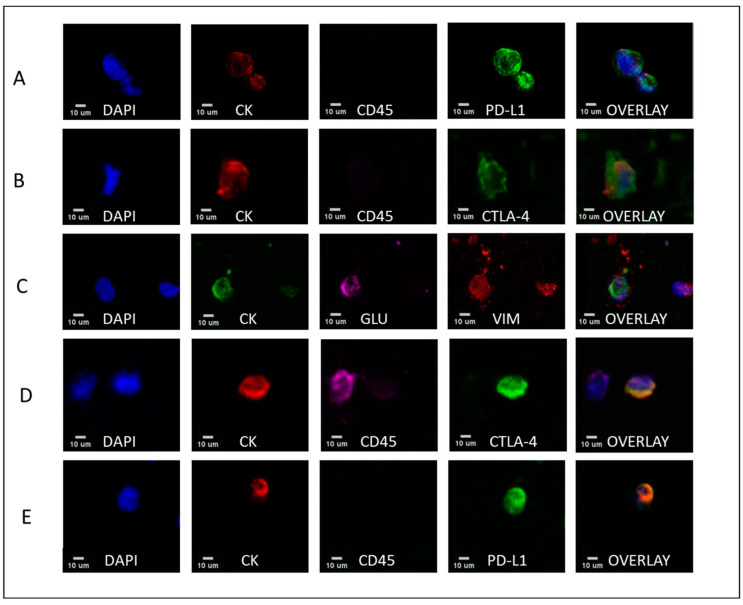
PD-L1, CTLA-4, GLU, and VIM expression in TNBC and luminal patients’ CTCs. The first column represents nuclei stained with DAPI, the second column represents cells expressing CK, the third cells expressing CD45 and GLU, the fourth cells expressing either PD-L1, CTLA-4, or VIM, while the fifth represents the overlay of the four channels. Representative panels of TNBC CTCs with (**A**) expression of PD-L1; (**B**) CTLA-4; and (**C**) expression of GLU and VIM. (**D**) Expression of CTLA-4 and (**E**) PD-L1 in luminal patients’ samples. Scale bars = 10 μm.

**Figure 4 cancers-15-01974-f004:**
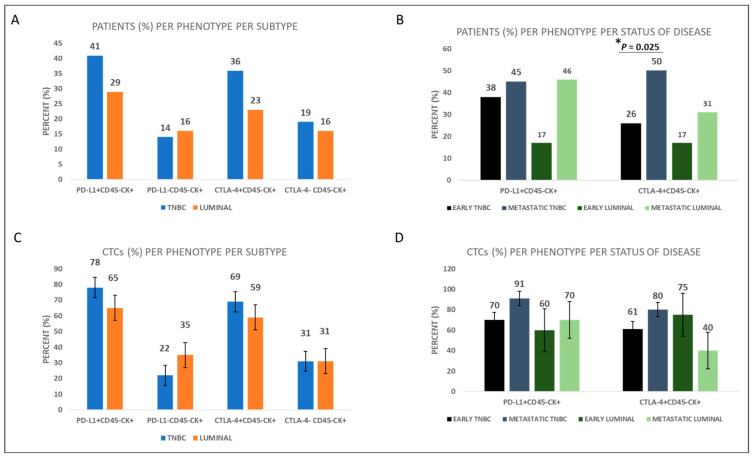
PD-L1 and CTLA-4 expression in BC patients’ CTCs. (**A**) Percentage of TNBC and luminal patients with PD-L1^+^CD45^−^CK^+^, PD-L1^−^CD45^−^CK^+^, CTLA-4^+^CD45^−^CK^+^, and CTLA-4^−^CD45^−^CK^+^ phenotypes. (**B**) Percentage of early and metastatic BC patients with the corresponding CTC phenotypes (* *p* = 0.025). (**C**) Percentage of CTCs with PD-L1^+^CD45^−^CK^+^, PD-L1^−^CD45^−^CK^+^, CTLA-4^+^CD45^−^CK^+^, and CTLA-4^−^CD45^−^CK^+^ phenotypes in TNBC and luminal patients. (**D**) Percentage of CTCs with the corresponding phenotypes in early and metastatic BC patients.

**Figure 5 cancers-15-01974-f005:**
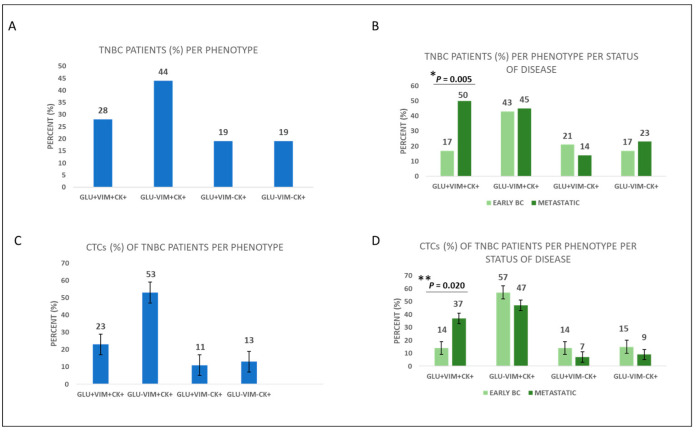
GLU and VIM expression in TNBC patients’ CTCs. (**A**) Percentage of TNBC patients with all the different phenotypes. (**B**) Percentage of early and metastatic TNBC patients with the corresponding CTC phenotypes (* *p* = 0.005). (**C**) Percentage of CTCs with all the corresponding phenotypes in TNBC patients. (**D**) Percentage of CTCs with all the corresponding phenotypes in early and metastatic TNBC patients (** *p* = 0.020).

**Figure 6 cancers-15-01974-f006:**
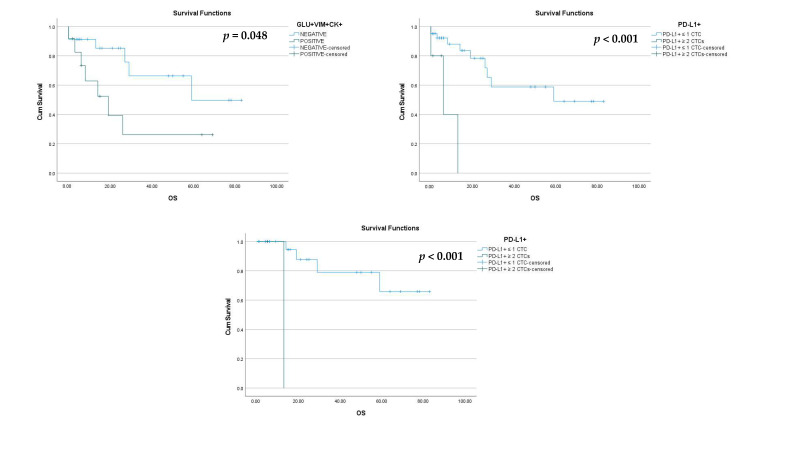
Estimate of OS of TNBC patients with respect to the expression of GLU, VIM, and PD-L1. (**A**) Kaplan–Meier survival curve for TNBC patients positive for GLU^+^VIM^+^CK^+^ phenotype (*p* = 0.048). (**B**) Kaplan–Meier survival curve for TNBC patients with two or more PD-L1+ CTCs compared to TNBC patients with one or no CTCs (*p* < 0.001). (**C**) Kaplan–Meier survival curve for early TNBC patients with two or more PD-L1+ CTCs compared to early TNBC patients with one or no CTCs (*p* < 0.001).

**Table 1 cancers-15-01974-t001:** Patients’ characteristics.

Early Disease (42 TNBC Patients)		Early Disease (18 Luminal Patients)		Metastatic Disease (22 TNBC Patients)		Metastatic Disease (13 Luminal Patients)	
**Age**		**Age**		**Age**		**Age**	
Median, range	58 (34–79)	Median, range	44 (41–59)	Median, range	58 (45–83)	Median, range	67 (50–79)
**Menopausal status**		**Menopausal status**		**Menopausal status**		**Menopausal status**	
Premenopausal	8 (19)	Premenopausal	4 (22)	Premenopausal	2 (9)	Premenopausal	2 (15)
Postmenopausal	18 (43)	Postmenopausal	9 (50)	Postmenopausal	9 (41)	Postmenopausal	8 (62)
Unknown	16 (38)	Unknown	5 (28)	Unknown	11 (50)	Unknown	3 (23)
**Tumor size**		**Tumor size**		**Disease sites**		**Disease sites**	
pT1	6 (14)	pT1	3 (17)	1	4 (18)	1	2 (15)
pT2	12 (29)	pT2	3 (17)	2	1 (4)	2	0 (0)
pT3	1 (2)	pT3	2 (11)	3	1 (5)	3	0 (0)
Unknown	23 (55)	Unknown	10 (55)	≥4	1 (5)	≥4	0 (0)
				Unknown	15 (68)	Unknown	11 (85)
**Histologic grade**		**Histologic grade**		**Primary breast cancer**		**Primary breast cancer**	
Grade 1	2 (5)	Grade 1	0 (0)	Adjuvant	2 (9)	Adjuvant	5 (38)
Grade 2	5 (12)	Grade 2	3 (17)	Metastatic	5 (23)	Metastatic	1 (8)
Grade 3	12 (28)	Grade 3	4 (22)	Unknown	15 (68)	Unknown	7 (54)
Unknown	23 (55)	Unknown	11 (61)				
**ER/PR tumor status**		**ER/PR tumor status**		**ER/PR tumor status**		**ER/PR tumor status**	
Positive	0 (0)	Positive	8 (44)	Positive	0 (0)	Positive	5 (38)
Negative	42 (100)	Negative	0 (0)	Negative	22 (100)	Negative	0 (0)
Unknown	0 (0)	Unknown	10 (56)	Unknown	0 (0)	Unknown	8 (62)
**HER2 tumor status**		**HER2 tumor status**		**HER2 tumor status**		**HER2 tumor status**	
Positive	0 (0)	Positive	0 (0)	Positive	0 (0)	Positive	0 (0)
Negative	42 (100)	Negative	7 (39)	Negative	22 (100)	Negative	5 (38)
Unknown	0 (0)	Unknown	11 (61)	Unknown	0 (0)	Unknown	8 (62)

## Data Availability

Data presented in this study are available upon request from the corresponding author.

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
