# Peer review of "Immune Checkpoint and EMT-Related Molecules in Circulating Tumor Cells (CTCs) from Triple Negative Breast Cancer Patients and Their Clinical Impact"

_cancers, 2023, doi:10.3390/cancers15071974_

Round 1
Reviewer 1 Report
In this manuscript, the authors investigate the expression of PD-L1. CTLA-4. GLU and VIM in CTCs of TNBC patients, and the authors concluded that these markers are relevant to TNBC patients’ outcomes. The authors just show a few results of these four markers in TNBC patients, not showing the difference between TNBC and Luminal patients. I think the current manuscript is not available for publication in Cancers.
Specific comments:
1. In Figure 1, the authors analyzed the PD-L1 and CTLA-4 expression in TNBC patients’ CTCs, whether the significant expression difference of PD-L1 and CTLA-4 was detected in early BC and metastatic patients? How about the P-value analysis?
2. In figure 2. 1) The authors show the expression of PD-L1. CTLA-4. GLU and VIM in CTCs of TNBC patients. How about the expression of these four markers in Luminal patients? 2) the authors did not show the details about the picture, such as, how many patients were analyzed? How many cells were analyzed in each patient? 3) the author should add the bar for each picture.
3. In figure 3, the authors should add the P-value analysis between TNBC and Luminal.
4. In figure S1, 1) The authors show the expression of PD-L1. CTLA-4. GLU and VIM in MDAMB231 cells. How about the expression of these four markers in MCF7 cells? 2) the authors did not show how many cells were analyzed? 3) the author should add the bar for each picture.
Author Response
REVIEWER 1:
In this manuscript, the authors investigate the expression of PD-L1. CTLA-4. GLU and VIM in CTCs of TNBC patients, and the authors concluded that these markers are relevant to TNBC patients’ outcomes. The author. just show a few results of these four markers in TNBC patients, not showing the difference between TNBC and Luminal patients. I think the current manuscript is not available for publication in Cancers.
Specific comments:
- In Figure 1, the authors analyzed the PD-L1 and CTLA-4 expression in TNBC patients’ CTCs, whether the significant expression difference of PD-L1 and CTLA-4 was detected in early BC and metastatic patients? How about the P-value analysis?
We thank the reviewer for the comment. Statistical analysis was performed for all the comparisons between different phenotypes. However, in the text, we included only the p-values of the statistically significant results.
In the revised version we added 12 more TNBC patients whose analysis was recently completed. New data confirmed the previous results, however, the new analysis showed that the expression of phenotype (CTLA-4+CD45-CK+) was significantly higher (p = 0.025) in metastatic compared to early TNBC patients (New Figure 2B) and the frequency of CTCs belonging to the phenotype (CTLA-4+CD45-CK+) was also higher (p= 0.024) in metastatic compared to early TNBC patients (New Figure 2C). In addition to these, the frequency of the phenotype (PD-L1-CD45-CK+) was higher in early compared to metastatic TNBC patients (p= 0.049). We have now included a table that demonstrates all the p-values from the conducted statistical analysis as supplementary (Table S1).
- In figure 2. 1) The authors show the expression of PD-L1. CTLA-4. GLU and VIM in CTCs of TNBC patients. How about the expression of these four markers in Luminal patients? 2) the authors did not show the details about the picture, such as, how many patients were analyzed? How many cells were analyzed in each patient? 3) the author should add the bar for each picture.
We thank the reviewer for this very valid point. We have now included in figure 2, (which is now figure 3), the expression of PD-L1 and CTLA-4 in Luminal patients. We have also added a scale bar in each photo. Luminal patients have not been stained for GLU and VIM since this analysis has already been conducted in our previous study (Kallergi et al., 2018). Furthermore, our main goal in this manuscript was the characterization of TNBC patients’ CTCs. In the revised manuscript we have also added a supplementary table (Table S2) showing the total number of patients analyzed per subtype, and the number of CTCs belonging to distinct phenotypes in every patient.
- In figure 3, the authors should add the P-value analysis between TNBC and Luminal.
In the manuscript, we only reported p-values ​​of statistically significant results. Therefore, there are no statistically significant p-values in figure 3, (which is now figure 4). We have now included, according to the reviewer’s comment, a table, demonstrating all the p-values from every distinct statistical analysis as supplementary (Table S1).
- In figure S1, 1) The authors show the expression of PD-L1. CTLA-4. GLU and VIM in MDAMB231 cells. How about the expression of these four markers in MCF7 cells? 2) the authors did not show how many cells were analyzed? 3) the author should add the bar for each picture.
According to the reviewer’s comment, we have now included in figure S1 the expression of PD-L1 and CTLA-4 in MCF7 cells and added a scale bar to each photo. MCF7 cells were not stained for GLU and VIM since this staining in the luminal subtype was examined in our previous study (Kallergi et al., 2018). Furthermore, details for the number of the analyzed cells are now described in the legend of supplementary figure 1 (page 14, line 499).
Sincerely,
Galatea Kallergi
Reviewer 2 Report
This manuscript is well written. In this study authors investigated the expression of immune checkpoint proteins and EMT markers in TNBC patients CTCs and demonstrated their relationship with clinical outcomes. However, I am little confused about some of the data. Why is it percentage of patients per phenotype or subtypes plots when combined is not tallying up to 100%. (For example: (PD-L1+CD45-CK+) + (PD-L1-CD45-CK+) != 100%)
Minor edits:
p value has to be added for all CTCs (%) of TNBC Patients per phenotype plots.
OS survival curves needs to be more clear. It is difficult to read with current font sizes.
Author Response
This manuscript is well written. In this study authors investigated the expression of immune checkpoint proteins and EMT markers in TNBC patients CTCs and demonstrated their relationship with clinical outcomes. However, I am little confused about some of the data. Why is it percentage of patients per phenotype or subtypes plots when combined is not tallying up to 100%. (For example: (PD-L1+CD45-CK+) + (PD-L1-CD45-CK+) != 100%)
We thank the Reviewer for the comment. The subtypes of patients per phenotypes when combined do not reach up to 100% because it is possible for a patient to harbor more than one type of CTCs in his blood, therefore it is possible to belong to different subgroups.
Minor edits:
p value has to be added for all CTCs (%) of TNBC Patients per phenotype plots.
In the manuscript, we only reported p-values ​​of statistically significant results. Therefore, there are no statistically significant p-values in the figures, however, we have now included, according to the reviewer’s comment, a table, demonstrating all the p-values from every distinct statistical analysis as supplementary (Table S1).
OS survival curves needs to be more clear. It is difficult to read with current font sizes.
We thank the reviewer for the comment. We have now increased the size of the fonts in the survival curves.
Sincerely,
Galatea Kallergi
Reviewer 3 Report
The study is well done, the material is large enough and the methods look reliable. The study is based on extensive and not recent literature, gives some new information and this warrants its publication.
Although several reviews about the diagnostics of breast cancer have been already published, the discussion on the markers of breast carcinoma in this paper seems to be original. However I have the following suggestions/comments and hope the authors can address them in the review.
Minor revision
1. Some authors showed that the ADH/ALDH activities are lower in tumor cells than in normal parenchyma, suggesting that isoenzymes of ADH may play an important role in carcinogenesis. Among all tested classes of ADH isoenzymes, only class I had higher activity in the serum of patients with breast cancer in stage IV:
Jelski W et al.,
The activity of class I, II, III and IV alcohol dehydrogenase isoenzymes and aldehyde dehydrogenase in breast cancer. Clin Exp Med (2006) 6:89–93
Jelski W et al.,
Activity of Alcohol Dehydrogenase (ADH) Isoenzymes and Aldehyde Dehydrogenase (ALDH) in the Sera of Patients with Breast Cancer.
Journal of Clinical Laboratory Analysis (2006), 20: 105-108
Please discuss (5-6 sentences)
Author Response
The study is well done, the material is large enough and the methods look reliable. The study is based on extensive and not recent literature, gives some new information and this warrants its publication.
Although several reviews about the diagnostics of breast cancer have been already published, the discussion on the markers of breast carcinoma in this paper seems to be original. However, I have the following suggestions/comments and hope the authors can address them in the review.
Minor revision
- Some authors showed that the ADH/ALDH activities are lower in tumor cells than in normal parenchyma, suggesting that isoenzymes of ADH may play an important role in carcinogenesis. Among all tested classes of ADH isoenzymes, only class I had higher activity in the serum of patients with breast cancer in stage IV:
Jelski W et al., The activity of class I, II, III and IV alcohol dehydrogenase isoenzymes and aldehyde dehydrogenase in breast cancer. Clin Exp Med (2006) 6:89–93
Jelski W et al.,
Activity of Alcohol Dehydrogenase (ADH) Isoenzymes and Aldehyde Dehydrogenase (ALDH) in the Sera of Patients with Breast Cancer. Journal of Clinical Laboratory Analysis (2006), 20: 105-108
Please discuss (5-6 sentences)
We thank the Reviewer for the comments. According to the reviewer’s suggestion, we have now discussed the activity of Alcohol Dehydrogenase (ADH) Isoenzymes and Aldehyde Dehydrogenase (ALDH) in the sera of patients with breast cancer (page 2, lines 83-87).
Reviewer 4 Report
The authors investigated the association between the expression of markers of epithelial-mesenchymal transition and immune checkpoint on CTCs and markers of immune checkpoint in triple negative breast cancer (TNBC) and found that PD-L1, CTLA-4, 39 GLU, and VIM constitute important biomarkers of TNBC associated with patient outcomes.
The results are very interesting and may lead to future treatments, and I think it is a useful study. However, there are some criticisms:
Major points
1. Since they are analyzing prognosis in a clinical retrospective study, please indicate the setting and clarify the indication and exclusion criteria in “Methods”. Also, please be sure to know what kind of case you analyzed with a flow chart in the results.
2. In the Method, specify the main endpoint (one in principle) and the secondary endpoint, and indicate the results accordingly.
3. Since the proportion of TNBC depends on the overall number of cases, I think authors should show the positivity rate of each marker in TNBC and that in non-TNBC cases.
Author Response
REVIEWER 3:
- Since they are analyzing prognosis in a clinical retrospective study, please indicate the setting and clarify the indication and exclusion criteria in “Methods”. Also, please be sure to know what kind of case you analyzed with a flow chart in the results.
We would like to thank the reviewer for the valuable comments. Exclusion criteria are now clarified in “Methods” (lines 150-153). Additionally, a flow chart has also been added in figure 1 (Figure 1A) which illustrates the total number of BC patients categorized in the two major subtypes (Luminal and TNBC) and their disease status (early and metastatic).
- In the Method, specify the main endpoint (one in principle) and the secondary endpoint, and indicate the results accordingly.
The primary endpoint of the study was the identification of CTCs belonging to the examined phenotypes. The secondary endpoint was the investigation of the potential clinical relevance of the observed biomarkers for TNBC patients’ outcomes. This clarification is now added to the revised manuscript (page 4, lines 156-158).
- Since the proportion of TNBC depends on the overall number of cases, I think authors should show the positivity rate of each marker in TNBC and that in non-TNBC cases.
We have now included in figure 1 the positivity rate of TNBC and luminal patients. We also include the positivity rate of early and metastatic TNBC and luminal patients. This clarification is now added to the revised manuscript (page 6, lines 222-228).
We remain at your disposal for any further information.
Sincerely,
Galatea Kallergi
Round 2
Reviewer 1 Report
I agree your response.
Reviewer 4 Report
The manuscript has nicely been revised .